# Designing and Evaluating a Portable UV-LED Vane Trap to Expedite Arthropod Biodiversity Discovery

**DOI:** 10.3390/insects15010021

**Published:** 2024-01-01

**Authors:** Seunghyun Lee, Michael C. Orr, Jinbae Seung, Yunho Yang, Zhehao Tian, Minhyeuk Lee, Jun-Hyung Tak, Seunghwan Lee, Ming Bai

**Affiliations:** 1Key Laboratory of Zoological Systematics and Evolution, Institute of Zoology, Chinese Academy of Sciences, Beijing 100101, China; chiyark@snu.ac.kr (S.L.);; 2Insect Biosystematics Laboratory, Department of Agricultural Biotechnology, Seoul National University, Seoul 08826, Republic of Korea; 3Research Institute for Agricultural and Life Sciences, Seoul National University, Seoul 08826, Republic of Korea; 4International College, University of Chinese Academy of Sciences, Beijing 100049, China; 5Entomologie, Staatliches Museum für Naturkunde Stuttgart, 70191 Stuttgart, Germany; 6Insect Pest Chemical Control Laboratory, Department of Agricultural Biotechnology, Seoul National University, Seoul 08826, Republic of Korea; 7National Institute of Agricultural Sciences, Wanju 55365, Republic of Korea

**Keywords:** collection method, passive trap, funnel light trap, DIY trap

## Abstract

**Simple Summary:**

Funnel light traps (FLTs) are effective in collecting insects as they are smaller, lighter, and easier to install than conventional light traps, and can be used in more remote areas; however, they can still be cost-prohibitive and logistically challenging, particularly in developing countries where much biodiversity remains to be discovered. In this study, we present a new, low-cost, UV-LED-based FLT trap that is both lightweight and easy to assemble and deploy in the field. Our field tests show that this trap is effective in collecting diverse insect samples in a range of habitats, including remote and difficult-to-access areas. Additionally, the trap’s modular design allows for easy customization and standardization, enabling comparability across sites and studies.

**Abstract:**

A novel design of a portable funnel light trap (PFLT) was presented for collecting insects in ecological studies. The trap consists of a compact plastic box equipped with a light source and power source, along with two plastic polypropylene interception vanes. The PFLT costs 18.3 USD per unit and weighs approximately 300 g. A maximum of six PFLT units can be packed in one medium-sized backpack (32 cm × 45 cm × 15 cm, 20 L), making it easier to set up multiple units in remote areas wherein biodiversity research is needed. The low cost and weight of the trap also allows for large-scale deployment. The design is customizable and can be easily manufactured to fit various research needs. To validate the PFLT’s efficacy in collecting insects across different habitat types, a series of field experiments were conducted in South Korea and Laos, where 37 trials were carried out. The PFLT successfully collected 7497 insects without experiencing battery issues or damage by rain or wind. Insect compositions and abundances differed across the three sampled habitat types: forests, grasslands, and watersides. This new FLT trap is an important tool for studying and protecting insect biodiversity, particularly in areas wherein conventional light traps cannot be deployed.

## 1. Introduction

Recent reports suggest that insects are in decline [1,2,3]; however, to protect species, we must know them, and this becomes increasingly difficult with growing species richness. Insects are the largest group of animals, accounting for approximately 70% of the total diversity of this planet [1,4]. Despite the description of more than one million insect species, millions more are expected and awaiting formal description [4,5,6]. The vast majority of these undiscovered and undescribed species exist in less-explored regions like tropical forests in the Neotropics, Afrotropics, and Indomalaya [7,8,9,10,11]. Additionally, even in better-known areas like developed countries, new species or species new to the region may often be found when collecting in less accessible areas like uninhabited islands, mountain peaks, or the center of large forests.

In remote areas, light trapping, inarguably one of the most effective methods for collecting insects, cannot easily be implemented [12]. Conventional active light trapping usually requires heavy, large equipment including an electric power generator, fragile mercury or metal halide and UV lamps, and the requisite connectors for them (Appendix A). Given these considerations, a well-paved road is almost a necessity, and these are lacking in many less-explored countries. In response to the limitations of conventional light traps, many funnel light traps (FLT) that attract nocturnal insects toward a light source, in which they run into vanes during flight and fall into collectors, have been developed. FLTs partially mitigate the shortcomings of conventional light traps because they are much lighter, smaller, and easier to install, and can be implemented in more remote areas than conventional light traps [12,13,14]. Consequently, FLTs generally yield high-quality and large quantities of insect samples with relatively little sampling effort [15,16], while also providing highly diverse collections [17].

Despite their effectiveness, FLTs are not always optimal research tools because of their cost and the logistics required to deploy them, both of which can be challenging in developing countries wherein much biodiversity remains to be discovered [8,18]. This is complicated by the fact that the microhabitat where the trap is installed strongly influences the species composition and diversity recovered, so there exists a statistical minimum number of traps for large-scale sampling. Unfortunately, no scientific paper or commercial company has provided an inexpensive, reuseable, and truly portable (i.e., light and small enough to carry more multiple traps per person) design, that is easily installable in multiple sites regardless of accessibility. Most importantly, all current designs remain bulky, too large to be mass-carried into inaccessible regions. In addition, commercial FLTs are too expensive ($200–$500 each, see [12,14]), while many DIY-style FLTs are crude and often unreliable (reviewed in [12]). Further, the use of non-standardized equipment hinders our ability to compare results across sites and studies, hindering our ability to assess and plan insect conservation actions [3,19].

Light-emitting diode (LED) technology has recently gained popularity as an attractant in light traps [20,21,22,23,24,25] due to its energy efficiency, cost-effectiveness, durability, and portability [24]. The wavelength spectrum typically ranges from 350 to 700 nm and different insect groups exhibit varying levels of attraction to specific wavelengths [20,25]. Notably, UV LEDs in the 350–400 nm range are widely recognized for their ability to attract a diverse array of insect groups, encompassing Lepidoptera [25], Diptera [22], Coleoptera, Hemiptera and Hymenoptera [24].

In this study, we developed a standard lightweight UV-LED-based FLT trap that is (1) portable via its modularity and small size, (2) easily assembled and deployed in the field, and (3) has high cost-performance. We conducted field tests of this trap in several regions with different habitat types (e.g., uninhabited islands, mountain peaks, a forest center, etc.) in neglected areas of South Korea and Laos. This trap is ideal for short-term overseas collection, especially in developing countries with limited infrastructure where it is almost impossible to operate conventional light traps or FLTs. We believe that this trap will accelerate species discovery and ultimately expand our knowledge of biodiversity in neglected regions.

## 2. Materials and Methods

### 2.1. Trap Design: Portable Funnel Light Trap (PFLT)

This new trap improves upon the common design of currently available FLT traps. Our trap consists of two parts: (1) the main box: a compact plastic box with a light source and power source installed inside, and (2) two plastic polypropylene interception vanes. The products and components (e.g., size, volume, etc.) specified in this paper are readily available in the authors countries (China and Republic of Korea) and could be manufactured without substantial difficulty; the trap may also be modified according to the specific needs of the end user, although by using a standardized design, one increases the comparability of methods across studies [19,26]. The estimated cost per trap and product details are summarized in Appendix A.

(1)The main box, including light and power sources (Figure 1A,B): The main body of the trap is a white plastic box of 230 mm × 160 mm × 50 mm that includes a lid that is as deep as the height of the box, fully covering the sides of it. The lid is used as a bottom collector, and the container, which is slightly smaller than the lid, has a light emitter at the top with UV LED straps that are connected to a portable mobile phone charger inside (which are widely accessible and can be as cheap as $5 USD). For the power source, we used a battery (portable power bank) with the following features: 3.7 v d.c., 10,000 mAh, DC 5V/2A, 68 mm × 136 mm × 15 mm in size and 209 g in weight. For the light source, we used a 5 m long waterproof UV LED strip with a light wavelength of 395–405 nm. Two 15 cm long strips with 10 LED units each were cut from the main strip and used. We attached the power source at the center and two strips on each side of the container. Starting from the battery, a 5V USB power connector, a switch, and two UV LED strips were connected by a wire and a snap connector.(2)Interception vanes (Figure 1C,D): White polypropylene danpla sheets of 5 mm thickness, which are widely used for corrugated shipping boxes because of their lightness and robustness, were used as vanes. Two square sheets of 300 mm × 300 mm were used for each trap. One sheet has a 5 mm × 150 mm longitudinal incision at the center, a 5 cm longitudinal cut at 70 mm from both corners at the bottom, and an 80 cm × 15 mm rectangular incision and a 5 cm longitudinal cut at 70 mm from both corners at the top. The other sheet has a 5 cm longitudinal cut at 35 mm from both corners at the top and the bottom, and a 150 cm × 15 mm rectangular incision followed by 5 mm × 130 mm longitudinal incision at the center, and has a 5 mm × 150 mm longitudinal incision at the top-center.(3)Assembly and installation (Figure 1E,G): First, the two vanes were assembled in a cross shape. The lower part of the box with the battery and UV LED attached was inserted into the upper grooves, and then the upper part of the box was inserted into the lower grooves. Finally, the collecting medium was poured into the bottom collector. Any type of liquid medium (e.g., ethanol, soap water, salt water, propylene glycol, etc.) can be used. The trap could then be placed on flat land, where it can stand firmly, based on the cross-shaped polypropylene danpla sheets. The UV LED strips and the portable charger were almost totally covered by the lower part of the plastic box, thus protected from natural hazards like rain or dew. If the trap needed to be installed at a higher point (trap height affects species composition, as shown in the results), its low weight enabled stable suspension via cable hanging.

**Figure 1 insects-15-00021-f001:**
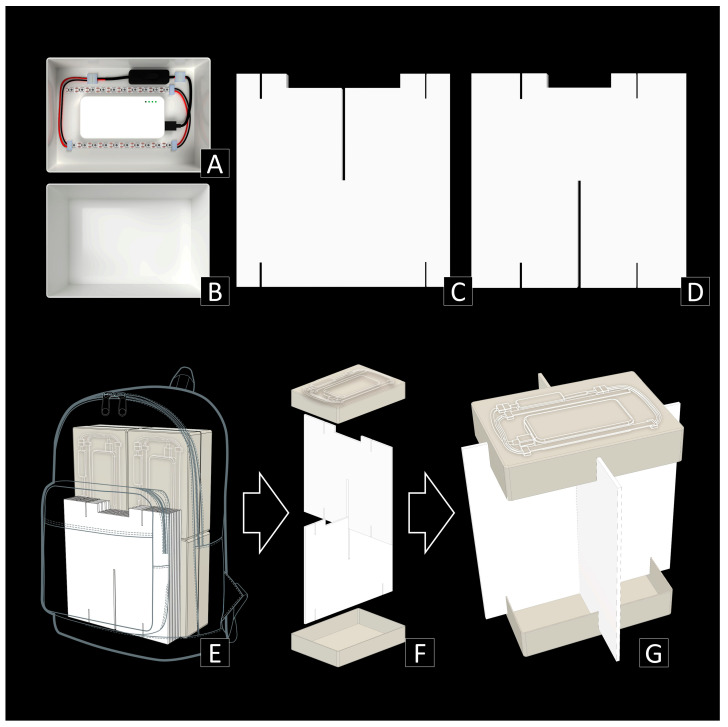
Trap design and utilization of PFLT. (**A**) The upper part of the main box, including light and power sources. (**B**) The lower part of the main box. (**C**) The upper interception vane. (**D**) The lower interception vanes. (**E**) PFLTs packed in a backpack. (**F**) Assembly. (**G**) Final installation.

### 2.2. Field Test

In 2022, a total of 37 trials were operated in different plots throughout Republic of Korea and in Southern Laos (Figure 2E), using soap water or ethanol as a preservation medium. The first 16 traps (Trial No. 1–16 in Appendix A) were installed in seven sites in Korea as pre-tests, mainly to test if the trap (i) successfully collects insects from diverse taxa, and (ii) can be easily installed and run overnight without having battery issues or being damaged by rain or wind. To test the PFLT under varying representative conditions, we tried to diversify the survey sites across latitude and longitude, habitat type, climate, vegetation, etc. (see Appendix A). After the pre-test, it was tested over several days in Jeju Island and Laos in August 2022, across three habitat types (forests, grasslands, watersides; Trial No. 17–37 in Appendix A). We compared the community composition of trapped insect orders from different habitat types when all the other conditions (e.g., temperature, moon phase, moisture, etc.) were similar. The traps were all installed in the afternoon while day collection took place, and were then checked the next morning. All collected insects were identified to order and counted in the field, partly sorted out, and stored in the Insect Biosystematics Lab of Seoul National University.

## 3. Results

The PFLT cost only USD 18.3 per unit (Appendix A), weighing approximately 300 g and taking up little space (23 cm × 16 cm × 5 cm for the main box, 30 cm × 30 cm × 1 cm for the vanes). A maximum of six PFLTs could be packed in one medium-sized backpack (32 cm × 45 cm × 15 cm, 20 L) and the total weight including collecting fluid was under 5 kg (Figure 1E). This compact size and light weight enabled us to install PFLTs in the center of the forest, where active light traps or other FLTs cannot be installed. Further, the PFLT outperformed in all five factors (cost, robustness, ease of installment, weight, and volume) compared to other existing FLT-style traps (Appendix A). During 37 independent operations including 16 pre-tests, not a single trap was broken or ran out of battery, despite operation in various sites with different habitat types and weather conditions. The time taken for installment was less than one minute per unit, and no trap operators reported issues in their use following minimal (>5-min) instructional demonstrations.

In total, 7497 insects from 12 insect orders were collected during the 37 trials (Archaeognatha, Blattodea, Coleoptera, Dermaptera, Diptera, Ephemeroptera, Hemiptera, Hymenoptera, Lepidoptera, Neuroptera, Orthoptera, Trichoptera). Overall, Lepidoptera was the most abundant, with 3494 specimens, followed by Coleoptera (1876 specimens), and Diptera (840 specimens). Across collections, the total abundance of insects and proportion of major insect orders differed, likely reflecting local conditions and faunas. For example, Coleoptera is clearly the most abundant taxa in all habitat types in South Korea, whereas it was the least abundant one in all habitat types in Laos (Figure 3). The number of collected insect orders, habitat type, GPS, date, weather, and remarks are summarized in Appendix A.

In the habitat type test, we found the total abundance of insects and the proportion of some major insect orders to differ by both region and habitat type (Figure 3 and Appendix A). In Laos, Lepidoptera was distinctly more abundant in the forest than other two habitat types. At the waterside, the abundances of Lepidoptera, Diptera, Hymenoptera, and Trichoptera were similar. Not surprisingly, Orthoptera were distinctly less abundant in forests than other habitat types, while Trichoptera were markedly more abundant in waterside habitats than other habitat types in both sites, which is expected, given their lifestyles. In Jeju island, the composition of insect orders across the three habitat types was found to be similar overall. However, notable differences were observed in the higher abundance of Coleoptera and Trichoptera in waterside habitats and Hemiptera in grassland habitats.

## 4. Discussion

The newly designed PFLT is low-cost, compact, light, and easy to assemble and disassemble, making it ideal for many types of biodiversity research, such as short-term collection, long-term biodiversity assessment, low-budget research, and even outreach and education, across a wider array of settings than previously possible. This is due in large part to its superior portability and cost-effectiveness.

A major advantage of the PFLT is its versatility in challenging field conditions, particularly in developing countries where access to electricity or stable road conditions may be limited [7,8,9,10,11]. Furthermore, insect surveys in these neglected areas have traditionally focused on tropical rainforests, with other habitats being largely neglected [27]. These features of PFLT also allow for the exploration of underexplored habitats, such as mangroves [27], alpine vegetation [28], and highland aquatic habitats [29] where new species are more likely to be discovered. Additionally, the PFLT can be used to sample insects at different heights, including above the canopy, as the composition, diversity, and abundance of insects can vary based on height [30,31,32]. The PFLT is designed to be lightweight and easy to install on tree branches, with a total weight of less than 1kg, as illustrated in Figure 3C,D. While our field tests did not reveal any vulnerability to wind, the design of the bottom collector can be customized to meet the specific needs of the end-user as needed (see the flexibility section).

Another main benefit of the PFLT design is to enable the comparability and standardization of collecting efforts, which is necessary for meaningful comparisons within and across sites [19,26]. For quantitative biodiversity evaluation, multiple traps installed in distant sites are required, and PFLTs can be adopted, especially for short-term surveys. Given that one unit can be installed in less than one minute in the field and many traps can be packed in a normal backpack (Figure 1E), it is theoretically possible to install dozens of traps within an hour. This can be achieved well ahead of the desired sampling period, as in our trials, the lights lasted over fifteen hours (also making this an ideal activity for complementary collecting mixed with other methods like netting, etc.). Along with the fact that the PFLT showed distinct taxa compositions between different habitat types, operating the PFLT for biodiversity assessment may provide more accurate measures of insect community abundance. This would enable more sophisticated sampling to study insect abundance and richness according to the canopy structure, height, different landscapes, or habitat types.

The biggest problem we encountered during the field test was the degradation of specimens of certain taxa. We used soapy water and ethanol as killing agents, but this greatly degraded the sample condition of some moths, causing scales to fall off and affecting the downstream preservation and identification of the specimens (Appendix A). The choice of a proper killing agent is crucial when preserving insects, especially when the end-user’s research focus is on morphology. For example, killing agents like chloroform or ethyl acetate should be used for purely morphological studies of difficult morphology such as moth wing scales, because they are better in preserving the quality of the specimen. In contrast, if the end-user’s research focus is on extracting DNA from specimens, ethanol should be used, as it preserves DNA well. In addition to the research focus, it is also important to consider other factors when choosing a killing agent. For example, if the research is being conducted in a remote location, a killing agent that is easy to transport and store should be chosen.

Likewise, the details of the PFLT (e.g., light source, size, collecting medium, power source, etc.) can be easily modified depending on the needs of the end user. For example, changing to light sources other than 295–400 nm UV-C can greatly affect the proportion of sampled insect orders. as different wavelengths of light attract different insects [22,23] and changing the plastic vane color could influence which insects are most attracted [33]. In addition, an ADUINO controller can be added to limit the time within which it is operated, to track the number of insects collected in a certain period within a day, or just to save power, thereby enabling multiple-day collections during specific periods daily (e.g., at night only).

Because another main purpose of designing PFLT is to provide a low-cost option both for researchers and the public, we used readily available materials to increase the likelihood that the trap can be built by more people, making it more widely accessible. All the components we used are easily obtainable via online stores; it is thus easy to replace parts when mechanical issues are encountered. This will increase average trap life, and the cost of maintenance will be lower.

## 5. Conclusions

In conclusion, the PFLT is an innovative and low-cost solution for collecting insects in challenging field conditions, particularly in developing countries wherein access to electricity or passable roads may be limited. Its high portability and cost-effectiveness make it ideal for biodiversity research across a wider array of settings than previously possible. Furthermore, PFLTs allow for the exploration of underexplored habitat types, such as mangroves and roadless mountain tops, where new species are more likely to be discovered. Additionally, they allow for the standardization of collecting efforts, which is necessary for meaningful comparison within and across sites, such as different heights of trees or different locations. The trap can be customized to meet the specific needs of end-users, but the development of a commercially available version of the trap that incorporates all these advantages is necessary to achieve greater standardization, sophistication, and durability, rather than relying on DIY versions. We believe that this kind of improvement in trapping methods will enable us to better uncover more species and assess diversity and abundance in different locations, ultimately leading to better baseline information on which species live where, and which species might be at risk.

## Figures and Tables

**Figure 2 insects-15-00021-f002:**
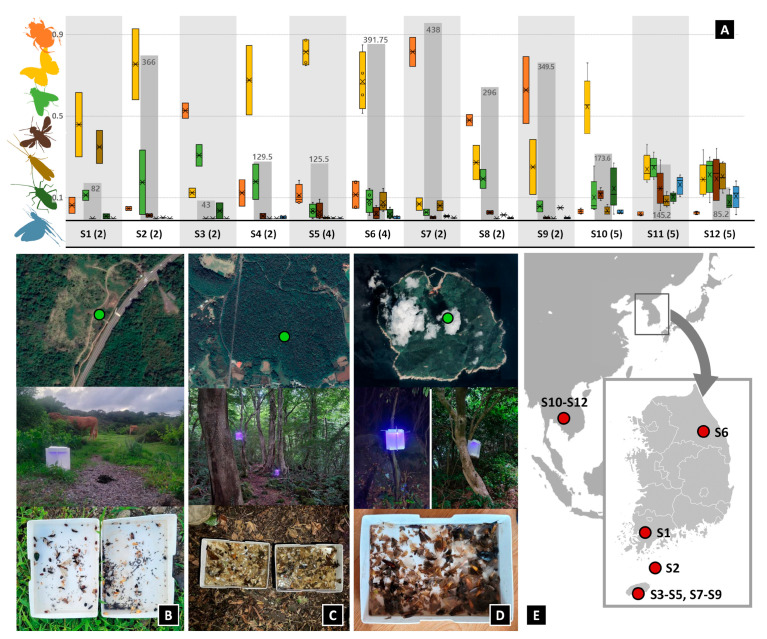
(**A**) Composition of collected insect orders (color bars) and total abundance (grey bars in the background). The color of the bars corresponds to the color of the insect order silhouettes on the left. The “o” marks in the colored bars represent data points between the lower whisker line and the upper whisker line, while the “x” indicates the mean marker for the selected series. The code starting with “S” corresponds to the survey site in the Appendix A, and the number in parentheses indicates the number of surveys conducted at the site. The results of the seven most abundant orders are shown. The numbers near the gray bar represent the average number of insects collected in one trap per day in Figure 2A. (**B**) Satellite image of survey site S8, with installed PFLT and collected samples. (**C**) Satellite image of survey site S5, with installed PFLT and collected samples. (**D**) Satellite image of survey site S2, with installed PFLT and collected samples. (**E**) Map of survey sites.

**Figure 3 insects-15-00021-f003:**
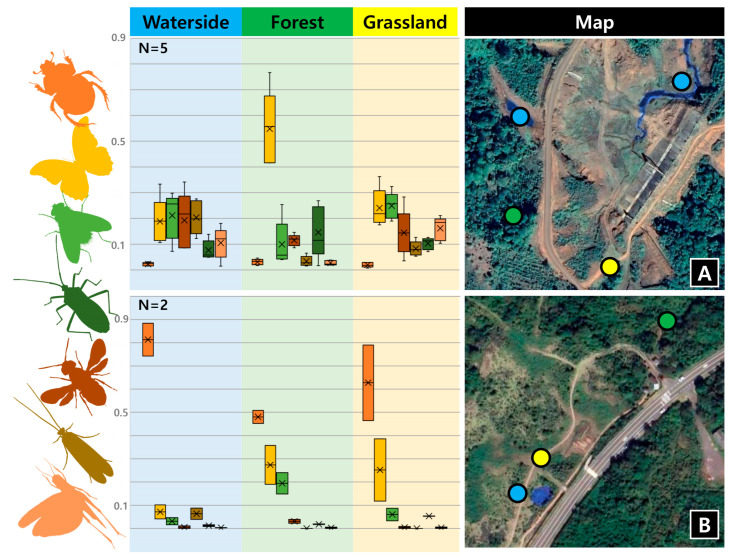
Insect order composition by habitat type. N = Number of repetitions. (**A**) Composition of insect orders by habitat types at survey sites 10–12 in Laos. (**B**) Composition of insect orders by habitat types at survey sites 7–9 in Korea. The color of the bars corresponds to the color of the insect order silhouette on the left. The numbers on the map correspond to the survey site in the Appendix A. The result of the seven most abundant orders are shown.

## Data Availability

The data presented in this study are available in the article and Appendix A.

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
