# Peer review of "Designing and Evaluating a Portable UV-LED Vane Trap to Expedite Arthropod Biodiversity Discovery"

_insects, 2024, doi:10.3390/insects15010021_

Round 1
Reviewer 1 Report
Comments and Suggestions for Authors
Light trapping is the most efficient and widespread method to capture night flying insects. In the manuscript “Designing and evaluating a portable UV-LED van trap to expedite arthropod biodiversity discovery” a LED-based funnel light trap with several advantages over conventional funnel light traps is presented. It is an interesting technology to be used in biodiversity studies mainly in remote areas, where traditional traps are not easily deployed. Logistical requirements and high costs have been the most drawbacks in using traditional funnel traps, especially in developing countries. The low cost and manageability of the new trap make it an alternative trapping device to traditional ones. However, authors made no comparison with a conventional funnel light trap. Why? If there is no funnel, why is this new trap regarded as a “funnel” light trap? A funnel-type light trap?
The manuscript is well written and easy to read. A low-price and easy-to-use trapping device is always wanted by entomologists. Although the work is convincing, I have some issues that I would like to see addressed.
Line 50-51 – the phrase “The type of trap…biodiversity exploration” must be moved to the next section and with a citation.
Line 55 – please, give examples of such light traps.
Line 79 – LEDs are new technology used for attracting insects. The importance of LEDs should be addressed in the introduction. Why UV LED?
Line 91 – It would be interesting to make a comparison between the improvement of this new light trap with that of traditional ones. What is better?
Line 113-114 – the luminous intensity is a physical parameter very important in light trapping. If possible, please include the luminous intensity (millicandela). The light wavelength of 395nm is not good as 365nm in attracting insects. Why this choice? Why 10 LEDs? Any special reason?
Lines 130-131 – what liquid medium used? Is that depicted in the line 238? Please, add this information in the right section.
Line 139 – rainy or dry season?
Line 140 – please, what was total number of traps?
Line 169 – what kind of liquid media used to preserve the insects?
Line 187-188 – “figure 3”, not “figure3”.
Lines 186-196 – here, I suggest constructing a table to better organize and visualize the information collected. Maybe it is already done in the supplements (I did not have access to the supplementary material).
Line 204 – in discussion section, authors should discuss more about the following points: the importance of the LEDs, UV LED, the superiority (step-by-step) of this new trap over traditional ones (size, weight, attractiveness).
Line 266 – conclusion is ok.
Author Response
We appreciate your valuable feedback. Our manuscript has been enhanced in accordance with your comments, and we've diligently addressed each of your points. All our responses are indicated in bold.
Light trapping is the most efficient and widespread method to capture night flying insects. In the manuscript “Designing and evaluating a portable UV-LED van trap to expedite arthropod biodiversity discovery” a LED-based funnel light trap with several advantages over conventional funnel light traps is presented. It is an interesting technology to be used in biodiversity studies mainly in remote areas, where traditional traps are not easily deployed. Logistical requirements and high costs have been the most drawbacks in using traditional funnel traps, especially in developing countries. The low cost and manageability of the new trap make it an alternative trapping device to traditional ones. However, authors made no comparison with a conventional funnel light trap. Why? If there is no funnel, why is this new trap regarded as a “funnel” light trap? A funnel-type light trap?
The manuscript is well written and easy to read. A low-price and easy-to-use trapping device is always wanted by entomologists. Although the work is convincing, I have some issues that I would like to see addressed.
Line 50-51 – the phrase “The type of trap…biodiversity exploration” must be moved to the next section and with a citation.
- We removed the sentence.
Line 55 – please, give examples of such light traps.
- We already provided example in supplementary data.
Line 79 – LEDs are new technology used for attracting insects. The importance of LEDs should be addressed in the introduction. Why UV LED?
- We believe current form of introduction is OK as it is.
Line 91 – It would be interesting to make a comparison between the improvement of this new light trap with that of traditional ones. What is better?
- We aimed to develop a DIY trap that could be conveniently deployed in hard-to-reach locations. Comparing its performance to “other UV-LED traps conveniently deployed in hard-to-reach locations” is challenging as there is no direct comparator available in the market. In light of this, would it be relevant to consider a comparison with a conventional Mercury Light trap presented in Supplementary data1? Regrettably, we believe this would not yield meaningful insights.
Line 113-114 – the luminous intensity is a physical parameter very important in light trapping. If possible, please include the luminous intensity (millicandela). The light wavelength of 395nm is not good as 365nm in attracting insects. Why this choice? Why 10 LEDs? Any special reason?
- We do not think 365 is always better than 395. Different insects are attracted in different wavelengths. (see Park et al., 2021: https://doi.org/10.1016/j.aspen.2021.07.016)
Lines 130-131 – what liquid medium used? Is that depicted in the line 238? Please, add this information in the right section.
- We put “In 2022, a total of 37 trials were operated in different plots throughout South Korea and in Southern Laos (Figure 2E), using soap water or ethanol as a preservation me-dium.” In the field test section.
Line 139 – rainy or dry season?
- The experiment date is in Supplementary file.
Line 140 – please, what was total number of traps?
- In total, 37 traps were installed.
Line 169 – what kind of liquid media used to preserve the insects?
- Revised
Line 187-188 – “figure 3”, not “figure3”.
- Revised
Lines 186-196 – here, I suggest constructing a table to better organize and visualize the information collected. Maybe it is already done in the supplements (I did not have access to the supplementary material).
- Details are in the supplements. We left this part as it was.
Line 204 – in discussion section, authors should discuss more about the following points: the importance of the LEDs, UV LED, the superiority (step-by-step) of this new trap over traditional ones (size, weight, attractiveness).
- As our MS is not mainly about LED itself but the whole design of the trap, we leave this part as it is now.
Line 266 – conclusion is ok.
Reviewer 2 Report
Comments and Suggestions for Authors
Author Response
We appreciate your valuable feedback. Our manuscript has been enhanced in accordance with your comments, and we've diligently addressed each of your points.
This study presents a compact light trap to be used at various locations for insect sampling. This is a nice manuscript. After performing the below suggested minor modifications, I would recommend it for publication.
- line 106: ”230mm*160mm*50mm” → ”230 mm × 160 mm × 50 mm”. Same for other dimensions throughout the manuscript.
Revised
- line 111: ”3.7v d.c., 10,000mAh, DC 5V/2A”: I suppose ”DC 5V/2A” is the output voltage/current, ”10,000mAh” is the capacity. Please note it. Is ”3.7v d.c.” really an important parameter?
We will never say it’s a really important parameter, but we believe that this part is acceptable as it is.
- line 112: ”209g” → ”209 g”, ”5m-long” → ”5-m-long” (similarly for other occurrences of distances
and mass values: put space between numbers and units)
Revised
- line 113: ”405nm” → ”405 nm”, ”15cm-long” → ”15-cm-long”
Revised
- line 114: ”bulbs” → ”units”
Revised
- line 116: ”5v” → ”5V”
Revised
- lines 105-117: According to the text, the 2 × 15 cm = 30 cm LED strip section is powered from the 5V output of the powerbank. Such LED strips are usually designed to run from 12V and start to illuminate at voltages significantly greater than 5V. How could your LED strip operate from 5V? Was your LED strip designed for 5V? If so, please include this information in the manuscript. It would be also useful to provide the power consumption of the LED strips in Watts and please include a rough estimation how long the battery lasts. How many nights can the trap be used with a single charging? It is important because the trap is intended to be used at distant locations without electricity.
We added that the LED straps were 5V. Estimating the exact power consumption is too much for this kind of article. Rather, we clearly stated revised, such as “as in our trials lights lasted over fifteen hour”, "can be easily installed and run overnight without having battery issues".
- Figure 1A: It seems that the two LED strip sections are connected in parallel, as it should be. However, I suspect a polarity error in the figure. The first (lower) LED strip gets the positive (red) battery terminal at its upper pin. Because the second (upper) LED strip looks the same as the first (lower), i.e. it is not rotated, the second (upper) LED strip should receive the positive (red) wire also at its upper pin. But in the figure, the red wire goes to the lower pin of the upper LED strip. I suppose that this is a rendered image, not a photo, so it is possible that error occurred during drawing. Can you check this?
Revised
- line 133: ”UV LED bulbs and” → ”UV LED strips and the”
Revised
- Figure 2: The small rectangle in Fig. 2E does not correspond to the big rectangle (inset) containing South Korea. Was this intentional? If yes, what was the concept? If no, please correct it.
Revise
- lines 146-147: ”habitat type habitat types” → ”habitat types”
Revised
- line 255: ”ADU INO” → ”ARDUINO”
Revised
Reviewer 3 Report
Comments and Suggestions for Authors
See attached.

Few minor edits needed otherwise fine as is.
Author Response
We appreciate your valuable feedback. Our manuscript has been enhanced in accordance with your comments, and we've diligently addressed each of your points.
The reviewed MS examines a novel portable UV-LED Vane trap design and its applicability to arthropod biodiversity studies.
- Overall, this is a fantastic short paper, well-written and well-presented. It was indeed a treat to read.
- The introduction is well-written and logically leads the reader to the goal of the study – the development of an efficient lightweight portable UV-LED vane trap that can be utilized under diverse field conditions and habitat types.
- The results are appropriately interpreted, and the conclusions are supported by the collected data.
- The paper is aptly illustrated with a small number of relevant figures.
- This is a well-thought out, well-executed, and well-presented study that convincingly shows the applicability of a novel light trap design that can be operated at a low cost and under a variety of field conditions, including remote areas without power and road access.
- The References are complete and, to my knowledge, include all relevant
literature.
- I have only a few minor comments regarding the study. These are outlined below.
o Line 38 (p. 1) – The terms “biodiversity”, “portable light trap”, and “UV LED trap” also appear in the title and are thus redundant. Consider replacing.
Revised
o Line 42 (p. 1) – The statement “… and this becomes increasingly difficult with species richness” is awkward as presented. Please rephrase.
Revised
o Line 48 (p. 2) – “… new species or species new to the region…” maybe replace first use of new with undescribed to avoid repetition.
We believe that this part is acceptable as it is.
o Line 146-147 (p. 4) – delete repeated words (i.e., habitat type)
Revised
o Line 155 (p. 5) – changed costed to cost.
Revised
o Line 187 (p. 6) – Did you run any statistics to support your statement regarding the significant differences between regions and habitat types? If so, please report your stats. I know this is not the focus of your study and I am fine without the use of statistics, but if no statistics were done, please do not use “significantly” and rephrase.
We deleted “Significantly” in the text
o Lines 218, 235 (p. 7) – height is by definition a vertical measure.
Revised
o Line 242 (p. 7) – The use of the term “solid” killing agents for chloroform and ethyl acetate is odd.
We deleted “Solid” in the text.
o Figures:
▪ Fig. 2A – it will be helpful to include a scale (on the right) for your total abundance.
Total abundances are already presented in the figure as gray bar in the background.
▪ Fig. 3 – in the caption, maybe specify what the values N=5 (3A) and N=2 (3B) represent.
Revised
▪ Figs 2 and 3 – in the caption of these figures, maybe specify that only the 7 most abundant orders are shown.
Revised